# Microscopical Variables and Tumor Inflammatory Microenvironment Do Not Modify Survival or Recurrence in Stage I-IIA Lung Adenocarcinomas

**DOI:** 10.3390/cancers15184542

**Published:** 2023-09-13

**Authors:** Andrea Dell’Amore, Alessandro Bonis, Luca Melan, Stefano Silvestrin, Giorgio Cannone, Fares Shamshoum, Alberto Zampieri, Federica Pezzuto, Fiorella Calabrese, Samuele Nicotra, Marco Schiavon, Eleonora Faccioli, Marco Mammana, Giovanni Maria Comacchio, Giulia Pasello, Federico Rea

**Affiliations:** 1Thoracic Surgery Unit, Department of Cardiac, Thoracic, Vascular Sciences and Public Health–DSCTV, University of Padova, 35128 Padova, Italysamuele.nicotra@aopd.veneto.it (S.N.); giovannimaria.comacchio@aopd.veneto.it (G.M.C.);; 2Pathology Unit, Department of Cardiac, Thoracic, Vascular Sciences and Public Health–DSCTV, University of Padova, 35128 Padova, Italy; federica.pezzuto@unipd.it (F.P.);; 3Oncology 2 Unit, Veneto Institute of Oncology IOV–IRCCS, 35128 Padova, Italy

**Keywords:** pathological variables, pathological score, tumor microenvironment, PD-L1, TILs, lung cancer, adenocarcinomas, early-stage lung adenocarcinoma

## Abstract

**Simple Summary:**

According to guidelines, resection remains a gold standard treatment in early-stage NSCLC. Because of the curative potential of surgery in these patients, microscopical and microenvironmental tumor processes in localized (N0) disease have been superseded for a long time and is a new emerging research field. Here, we investigated the influence of pathological variables and tumor immune environment in terms of survival and recurrence in resected adenocarcinomas staged I-IIA.

**Abstract:**

Microscopical predictors and Tumor Immune Microenvironment (TIME) have been studied less in early-stage NSCLC due to the curative intent of resection and the satisfactory survival rate achievable. Despite this, the emerging literature enforces the role of the immune system and microscopical predictors as prognostic variables in NSCLC and in adenocarcinomas (ADCs) as well. Here, we investigated whether cancer-related microscopical variables and TIME influence survival and recurrence in I-IIA ADCs. We retrospectively collected I-IIA ADCs treated (lobectomy or segmentectomy) at the University Hospital (Padova) between 2016 and 2022. We assigned to pathological variables a cumulative pathological score (PS) resulting as the sum of them. TIME was investigated as tumor-infiltrating lymphocytes (TILs < 11% or ≥11%) and PD-L1 considering its expression (<1% or ≥1%). Then, we compared survival and recurrence according to PS, histology, TILs and PD-L1. A total of 358 I-IIA ADCs met the inclusion criteria. The median PS grew from IA1 to IIA, indicating an increasing microscopical cancer activity. Except for the T-SUVmax, any pathological predictor seemed to be different between PD-L1 < 1% and ≥1%. Histology, PS, TILs and PD-L1 were unable to indicate a survival difference according to the Log-rank test (*p* = 0.37, *p* = 0.25, *p* = 0.41 and *p* = 0.23). Even the recurrence was non-significant (*p* = 0.90, *p* = 0.62, *p* = 0.97, *p* = 0.74). According to our findings, resection remains the best upfront treatment in I-IIA ADCs. Microscopical cancer activity grows from IA1 to IIA tumors, but it does not affect outcomes. These outcomes are also unmodified by TIME. Probably, microscopical cancer development and immune reaction against cancer are overwhelmed by an adequate R0-N0 resection.

## 1. Introduction

In recent decades, different pathological variables have been introduced and updated as survival or recurrence predictors in NSCLC [1]. This enhanced the scientific debate with a progressive and deeper development of the TNM classification and a new upgrade is expected to be released in 2024 [2].

The prognostic role of pathological issues (such as the spread through air spaces, vascular and perineural invasion, pleural involvement, tumor grading, histology or tumoral necrosis, mitoses and fibrosis) encompasses one of the three big areas of the modern cancer fields: (1) environmental exposure and ethnic characteristics, (2) microscopical features and (3) molecular cancer landscape. Across molecular and microscopical parameters, a fourth predictor has grown in the last few years: the Immune System (IS) and its microscopical related phenomenon, the Tumor Inflammatory Micro-Environment (TIME). According to the milestone theory of cancer environment, there are two different TIME patterns: the hot tumor with an enriched immune infiltrate and the cold tumor, a non-inflamed cancer with depleted immune infiltrating cells [3].

As a background, TIME has almost three related variables: the Programmed Death Ligand 1 (PD-L1), Tumor Infiltrating Lymphocytes (TILs) and Tumor Mutational Burden (TMB) [4]. The interaction of cancer with the PD-1 protein, expressed on the killer T-cells surface, induces an inhibitory reaction switching off the immune killing process [5]. This elicits an immune tolerance giving a sort of protection to the neoplasm that increases its malignant potential. Different international trials highlighted an improved Overall Survival (OS) and Disease-Free Survival (DFS) in locally advanced and advanced NSCLC (and recently also in the resectable disease) by targeting TIME either in adjuvant or neoadjuvant experiences [6,7,8,9,10,11,12,13]. In accordance with these comfortable results, a deeper comprehension of lung cancer biology and environment is warranted. 

As concerns Tumor Infiltrating Lymphocytes (TILs), since their introduction in melanoma [14], they were deeply investigated in terms of OS and DFS in different cancers, with a slight protective but inconclusive effect across metanalysis [15,16]. TILs and PD-L1 are different manifestations of the same phenomenon: TILs represent the IS response against cancer and PD-L1 is a cancer-related escape mechanism inducing TILs anergy [5,16]. TILs behavior in NSCLC (and ADCs as well) is not entirely decoded: they are synergic participants in the immune response against with other microscopic features (such as angioinvasion, pleural invasion or tumor necrosis) but different aspects need to be clarified yet.

Consequently, we aimed to explore whether the microscopical sum of variables, the PD-L1 expression or TILs influenced survival or recurrence in I-IIA early-stage resected ADCs.

## 2. Materials and Methods

We present a retrospective monocentric (Thoracic Unit, University of Padova) study that enrolled and operated (lobectomies or segmentectomies) upfront resectable adenocarcinomas (ADCs), between the 1 January 2016 and the 31 December 2022 in I-IIA stage.

This study was conceptualized and conducted in accordance with the Declaration of Helsinki and all patients gave written informed consent to adhere to our department’s research activity.

### 2.1. Inclusion and Exclusion Criteria

We selected primary adenocarcinomas of the lung (p-stage I-IIA) treated with anatomical resections (lobectomy or segmentectomy) and radical hilar and mediastinal lymphadenectomy. As a timeline, we included procedures performed between January 2016 and December 2022.

We excluded any benign pulmonary neoplasms or any tumor as a localization of metastases. We excluded other histology than adenocarcinoma, according to the findings of Ding and Colleagues [17], any p-stage IIB to IV or unresectable disease. We voluntarily avoided wedge-resections (atypical lung resections) or any resection which did not respect anatomical layers of the lung. We excluded any patient who received neoadjuvant treatments before surgery.

The inclusion of patients in I-IIA stage (based on the 8th AJCC TNM classification) was driven by the intention to highlight the results of those which received a radical surgical resection for a localized disease at pTNM evaluation R0-pN0 and did not receive adjuvant treatments but only radiological surveillance.

### 2.2. Data Collection and Management

Anamnestic schedule and preoperative examinations were collected in the archives. We registered: age, sex, the Body Mass Index (BMI), the smoking habitude and the Charlson Comorbidity Index (CCI). A preoperative pulmonary function test was routinely requested for the evaluation of One Lung Ventilation tolerance. We registered Forced Vital Capacity (FVC), Forced Expiratory Volume in one second (FEV1), alveolar carbon monoxide diffusion limit (DLCO/VA). We collected clinical staging investigations (cTNM), the surgical act.

### 2.3. Pathological Examination

Pathological data were obtained regarding the preoperative biopsy investigations (to register any preoperative diagnosis) and the analysis of the resected specimens (histotype and other histological characteristics of the lesion such as growth pattern, grading, number of mitoses, necrosis, lymphocytic infiltrate (TILs), STAS (Spread Through Air Spaces), vascular invasion, pleural invasion, perineural invasion, extent of resection margins and pathological TNM staging). The expression of PDL1 was also collected as percentage. Then, we dichotomized those expressing PD-L1 (≥1%) and those with <1% of PD-L1, in accordance with previous articles [18,19].

### 2.4. Pathological Score (PS)

We further assigned a score to pathological variables as shown in Table 1. The total score (PS) was obtained by the sum of each single variable. The PS was used to evaluate survival and recurrence, with a range of 0 to 9 points (Lower Score—LS) and 10 to 18 points (Higher Score—HS).

### 2.5. Statistical Analysis

Continuous variables were synthesized with median and interquartile range (IQR), categorical variables with absolute number and percentage frequency. Comparison between groups was investigated with Fisher’s exact test, Pearson’s chi-squared test or Wilcoxon’s test.

Survival analysis was plotted with the Kaplan–Meier estimator for Overall Survival (OS); Recurrence was investigated due to the Cumulative Hazard function. Curves were compared running a Log-rank test.

Analysis and graphs were obtained with Jamovi software (v2.3.21) [21] and R statistical software (v4.2.2—R Core Team 2022 [22]), with the ggplot2 and survival packages. Significance was set at *p* ≤ 0.05.

## 3. Results

Reviewing our surgical activity, 358 clinically resectable adenocarcinomas (ADCs) met the inclusion criteria. According to the 8th International Classification, our sample was mostly represented by p-stages IB and IA2 (176 cases—49% and 83 cases—23%). P-stage IIA has been reported in 15 cases—4.2%) (Figure 1).

### 3.1. Perioperative Evaluation

A total of 358 cases (179, 50% females and 179, 50% males) were treated (280 lobectomies or 78 segmentectomies) in the considered timeline and were consequently analyzed. They presented a median age of 70 years (IQR 63–75), with a diffuse former or active smoking habitude (259, 73%). The most representative comorbidities were hypertension (206, 58%), diabetes mellitus (38, 11%) and COPD (43, 12%). According to the Charlson Comorbidity Index (CCI), our sample presented a median score of 4 (IQR 3–5). Lung volumes were in median permissive to lobectomy. Trans alveolar diffusion of carbon monoxide was just slightly impaired (Table 2).

Preoperative enhanced chest CT-scan measured a median diameter of 20 mm (IQR 13–29); considering the clinical stage (cTNM), our sample was distributed as follows: 28 cT1a (7.8%), 136 cT1b (38%), 117 cT1c (33%), 49 cT2a (14%) and 28 cT2b (7.8%). Half of the patients (180, 50%) did not receive a preoperative conclusive diagnosis (due to bronchoscopy-assisted biopsy, CT-guided or US-guided transthoracic biopsy).

Peritumoral ground-glass opacity was identified in 123 cases (34%).

In 319 cases (89%), we performed a triportal Video Assisted Thoracic Surgery (VATS), in 28 (7.8%) interventions, we used a biportal VATS and in 6 cases (1.7%), we used a monoportal thoracoscopic approach. In 4 cases (1.1%), we used a daVinci^®^ Xi Robot-Assisted Thoracic Surgery.

We routinely performed the complete locoregional lymph nodes resection. Four VATS (1.1%) needed to be converted in thoracotomy due to bleeding hard adherences that could not be efficiently separated, maintaining the hilum under safe control (Table 2).

### 3.2. Pathological Data

A total of 279 specimens (78%) presented an acinar growth pattern, followed by lepidic (41, 11%) and solid (38, 10%). Other patterns were sporadic. In total, 258 neoplasms (73%) were G2, one fifth (79, 22%) were G3.

Spread Through Air Spaces (STAS) was expressed in 147 tumors (41%), generally in a limited form (135, 38%). Extensive STAS was rare (12, 3.4%). TILs infiltrated ADCs in 158 cases (45%). One fifth of specimens presented microscopical evidence of vascular invasion (65, 18%) and pleural invasion within the elastic membrane was diffused (PL1 and PL2) (194, 58%). Perineural involvement (5, 1.8%) was anecdotic. PD-L1 was expressed in 140 cases (47%) at least in a 1% of cells according to the TPS score. Other characters are available in Table 3.

Apparently, any pathological predictors seemed to be statistically different between the PD-L1 expressing and non-expressing groups. Although not significant, TILs were tendentially 10% more represented in PD-L1 > 1% neoplasms (63% vs. 54%) compared to stains not expressing PD-L1, which presented 46% of specimens with ≤10% of TILs (vs. 37% in PD-L1 > 1%) (Table 3).

The maximum Standard Uptake Value in resected PD-L1 expressing cancers was statistically higher (a median maximum Standard Uptake Value of 4 compared to 3.1 of PD-L1 < 1%, *p* = 0.01), indicating a higher glucose metabolism in PD-L1 > 1% ADCs (Table 4).

### 3.3. Survival and Recurrence According to the Pathological Score (PS)

We registered 267 cases (75%) of LS-ADCs and 91 cases (25%) of HS-ADCs. As expected, the median PS increased (non-linearly) from IA1 to IIA, accounting for a higher microscopical activity rate in IIA resected ADCs (Figure 2) compared to I. Median PS (IQR) in IA1, IA2, IA3, IB and IIA were, respectively, 6 (5–8), 7 (6–9), 7 (6–9), 8 (7–10) and 9 (7–10).

Considering survival, histology did not demonstrate a specifical subtype affecting survival (*p* = 0.37) (Figure 3A). As this premise, we investigated how the increase in PS could modify survival in our sample. Despite a graphically divergent survival trend, significance was far from being demonstrated (*p* = 0.25) and our patients presented a LS-ADCs survival comparable to HS-ADCs (Figure 3B). As survival was not significant, also recurrence was non relevant according to statistics considering the PS or histology (Figure 4A,B).

### 3.4. Immune Microenvironment and Survival

Giving results of the previous paragraph, we assumed that, in our experience, survival and recurrence were not strictly related to the number of pathological predictors expressed by the neoplasm.

With regard to TIME, we selected both PD-L1 and TILs, as they are different expressions of the same phenomenon: the activation of the IS against cancer. We separately investigated if a PD-L1 < 1% or a PD-L1 ≥ 1% and a lower (TILs < 11%) or a moderate/higher (≥11%) lymphocytes infiltration influenced our outcomes. Apparently, neither survival nor recurrence were related to TIME. In particular, a higher infiltration of TILs provided a comparable survival (*p* = 0.41) and recurrence (*p* = 0.97) in resected ADCs (Figure 5A,B). Moreover, no statistical significance or trend was available in I-IIA patients in terms of survival or recurrence considering PD-L1 (*p* = 0.23 and *p* = 0.74, respectively—Figure 6A and Figure 6B).

## 4. Discussion

In early-stage resectable ADCs, the role of microscopic variables or molecular mutations have been superseded for a long time, due to comfortable results of resected patients which achieved satisfactory survivals. Nevertheless, a deeper microscopical comprehension of cancer biology is providing new debates in NSCLC treatment. Our study emphasized the importance of resection in I-IIA ADCs but highlighted several aspects that need to be further discussed. According to our results, histology, PS and TIME were unrelated to survival or recurrence and the lack of a general statistical consistency in our data could be an unexpected, interesting result.

We can speculate that there could be two reasons that justify our results: first, it might be that a statistical difference between PS or TILs (and PD-L1 as well) needs a longer surveillance time to emerge. Nevertheless, another possible reason is that a radical N0 resection remains the best upfront treatment suitable for I-IIA ADCs. It must be considered that we included patients with the best p-stage available in terms of survival expectances.

Indeed, several considerations need to be underlined. A total of 75% of patients had a PS ≤ 9, indicating that early-stage ADCs have small, microscopical evidence of cancer activity. Considering that TNM classification is still predominantly related to the tumor size and less to specific microscopical pattern, the inconsistency of statistics in resectable I-IIA ADCs is comfortable: the TNM probably provides, as of today, an adequate classification for these patients. Notably, environmental, ethnic and molecular aspects were not investigated in our model, and it is possible that other important predictors are yet to be described. As an example, it was previously demonstrated as the spatial distribution of cancer cells and specific nuclear features analyzed by a mathematical software were able to correctly predict recurrence in different cohorts of ADCs in stage I-II [23].

Secondly, lymphocytes and PD-L1 expressing cancer cells have a still uncertain relation in the first part of the natural history of ADCs. As introduced with the cancer immunogram, TIME, TILs and Tumor Burden (TMB) play a convergent role with PD-L1 signaling to modulate immune response against the neoplasm [24,25]. This deeper relation between cancer and IS is intriguing. Notably, 140 patients (47%) expressed PD-L1 at least in 1% of analyzed cells. Moreover, 60% of resected ADCs presented more than 10% of Tumor-Infiltrating lymphocytes. Consequently, immune priming is expected to be an early phenomenon in cancer development, even if the interaction between TIME and ADC was inconsistent in this study.

As a premise, at least four critical points must be addressed on TILs. First, TILs were studied on several different neoplasms (such as in ovarian cancer and in colorectal neoplasm) but their specific contribution in NSCLC has not been largely investigated [15]. Moreover, Tumor Inflammatory Microenvironment (TIME) is extremely complicated, and it is still an innovative field in pre-clinic research [26]; consequently, clinical experiences are scarce. Thirdly, TIME heterogeneity could be responsible for inconsistency among different studies, providing incomparable results among similar research [27]. Finally, how squamous lung cancers (SqC) and ADCs have different behavior in terms of TILs-to-cancer interaction was addressed and they presented different subpopulations of TILs. Consequently, separate models should be provided [17]. As an example, regarding different histologies included in the same group, Schmidt et al. reported a non-significant difference in terms of OS in 321 NSCLC (stage I to III) stratified by PD-L1 status (*p* = 0.265), whereas survival in SqC demonstrated to be significantly improved (*p* = 0.04) [28]. Cooper et al. [29] evidenced a PD-L1 protective effect in squamous carcinomas but not in ADCs with an overexpression of PD-L1 (cut-off for overexpression was a PD-L1 higher than 50%).

In a previous article, in pN0 patients, the authors observed a non-significant better prognosis in high-expressing PD-L1 patients compared to low PD-L1 group. In pN1 and pN2 patients, a postoperative highly expressed PD-L1 was a worse prognostic variable [30].

A survival inconsistency in PD-L1 overexpressing NSCLC (higher than 50%) was previously reported by Zens and Colleagues but Kaplan–Meier results become significant when adopting the cut-off of 1% (*p* = 0.03), regardless histology (SqC 47.4%/ADCs 52.6%) [19].

The PD-L1 rating (Z score or different percentages) needs to be considered when comparing results in different studies. This is an important limitation that could bias the comparisons.

Our inclusion criteria (only I-IIA ADCs) aimed to reduce three important possible biases as much as possible: the inclusion of ADCs and SqCs, the role of neoadjuvant and adjuvant treatments and the role of an N1 disease. A lymph node involving cancer is by definition an escaped tumor and the complexity of the cancer microenvironment make comparisons extremely prone to bias.

Thus, there is an emerging probable ambivalent role of PD-L1: in locally advanced and advanced disease, it is certainly a worse predictor for survival. In our whole cohort of upfront resectable adenocarcinomas, it may not be conclusive. We suspect that the lack of results in IA1-IIA, if it really exists, can be explained: a radical surgical resection in the pN0 sample is likely to have resulted in a statistical inconsistency in terms of PD-L1 due to the complete removal of the tumor. Consequently, we speculate that, if present, any effect (positive or negative) of PD-L1 could be hidden by adequate surgical resection.

Regarding TILs, CD8+ were found to be significant in terms of OS and progression-free survival in a recent metanalysis that evaluated TILs in different neoplasms which included NSCLC [31]. In 136 untreated NSCLC (staged I to II), there was an increased survival in the PD-L1 expressing group (*p* = 0.044). Considering TILs, CD8+ provided an increased survival expectancy (*p* = 0.001). In the combined model, the CD8+/PD-L1neg group showed the highest survival probabilities while others presented a comparable survival rate. Notably, SqC histology accounted for 66.2% of the included cases [27]. Our cohort did not demonstrate any significance in terms of survival or recurrence on TILs. It is possible that the lymphocytes that infiltrated the tumor did not have enough time to make a significant difference. Therefore, it was previously reported that in seventy-nine patients (stage II–III, 65% ADCs) an immunoscore with the identification of TILs subpopulation was able to correctly influence overall survival, sustaining the hypothesis that shifting to locally advanced and advanced disease (escaped tumors), TIME in an important variable [32].

We were able to investigate only the percentage of TILs, but other parameters (such as the spatial distribution, TILs type and the TILs function) may modulate outcomes in cancers [18,26]. Therefore, it remains to be clarified how the pre-clinical evidence discussed above could be adopted as a routinary examination in the future. Advances in technology (such as machine-learning and radiomics) will provide innovative tools to translate pre-clinical findings into clinical assessment able to pivot the daily decision process [17,26,33].

We address some limitations: first, the retrospective and single-center structure of this study needs to be further confirmed in wider experiences. Secondly, we voluntarily included only adenocarcinomas. Consequently, this study could not be surely generalized for NSCLC. Moreover, we noted that we did not study the subtype of neoplasm entering lymphocytes despite a previous study disclaimed that NSCLC (in which 73% of specimens were ADCs) are generally highly enriched of CD8+ T-cells [18]. Consequently, we may suppose that our TILs were, at most, powered by CD8+ T-cells.

Despite this, the development of a routinary examination of lymphocytes sub-population is advisable. Finally, different TILs scoring and several experiences with both ADCs and other histology made it difficult to analyze results homogeneously.

Further and wider experiences need to be addressed to explore these emerging research fields, continuing to discover and to determine the role of tumor inflammatory microenvironment and microscopical factors in terms of survival and recurrence of early-stage resectable NSCLC.

## 5. Conclusions

Histology alone, the PS, PD-L1 and TILs failed to demonstrate a significant role in terms of survival or recurrence in IA1-IIA patients. As previously discussed, a radical surgical resection (R0 and N0) probably continues to provide, as of today, the best upfront treatment in early-stage ADCs, tempering the role of those variables. The most difficult step will be to overlap expensive pre-clinical findings to develop new technologies and predictors able to routinely process these insights in the daily clinical practice.

## Figures and Tables

**Figure 1 cancers-15-04542-f001:**
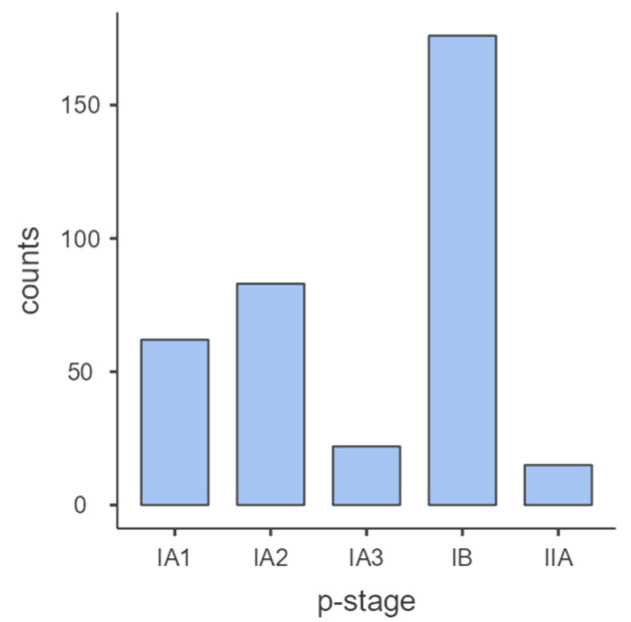
The p-stage distribution. P-stage IB was extremely represented.

**Figure 2 cancers-15-04542-f002:**
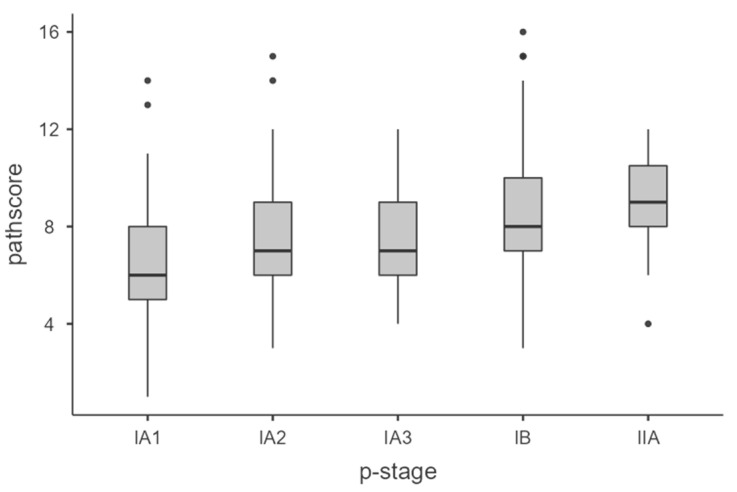
The median Pathological Score increased from IA1 to IIA. Each Box and Whisker Plot represents the distribution of PS in all the different p-stages. The black line in the box represents the median. Black points represent outliers.

**Figure 3 cancers-15-04542-f003:**
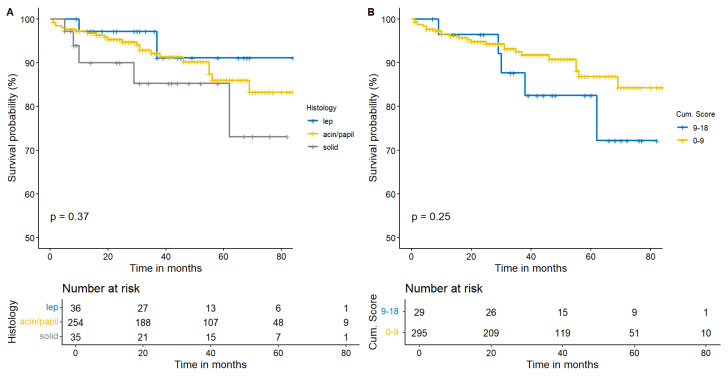
Kaplan–Meier estimates for survival demonstrated a non-significant survival difference among histology (**A**). Even in the PS model (HS-ADCs vs. LS-ADCs), survival was comparable (**B**).

**Figure 4 cancers-15-04542-f004:**
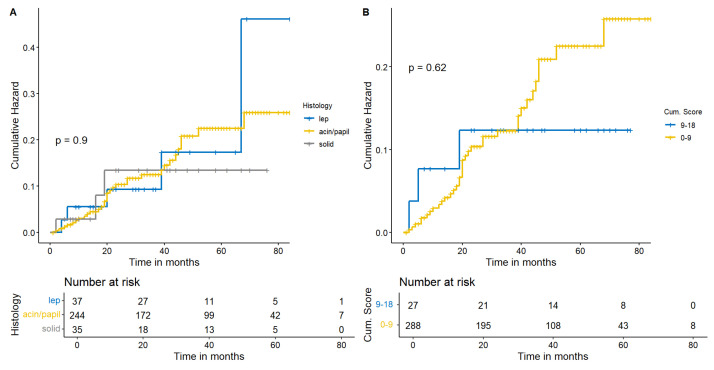
The Cumulative Hazard Function highlights a statistical inconsistency in terms of recurrence between different histologies (**A**). Moreover, HS-ADCs did not demonstrate to be related to an earlier recurrence compared to LS-ADCs in this model (**B**).

**Figure 5 cancers-15-04542-f005:**
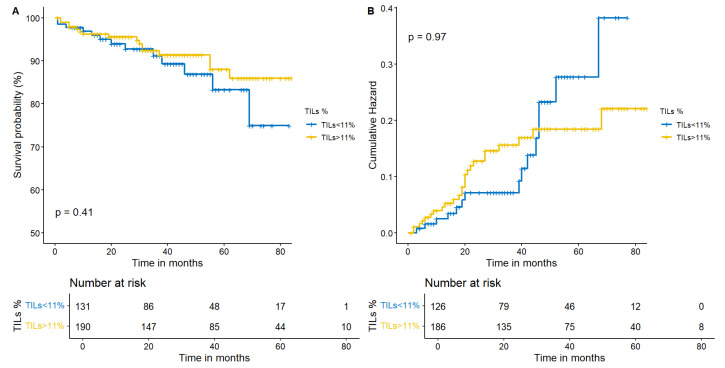
Kaplan–Meier estimates for survival and recurrence according to TILs. Neither survival (**A**) nor recurrence (**B**) were significantly modified by neoplasm entering lymphocytes according to the Log-Rank test.

**Figure 6 cancers-15-04542-f006:**
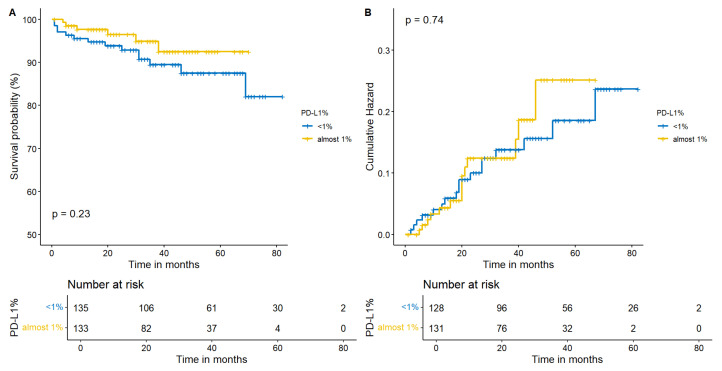
Kaplan–Meier estimates for survival and recurrence according to the PD-L1 status. No significance has been registered for survival (**A**) or in recurrence (**B**) in this model (PD-L1 < 1% or PD-L1 ≥ 1%).

**Table 1 cancers-15-04542-t001:** Pathological Score—PS. All predictors available in the pathological report with a double or triple outcome were included in the count.

Variable	P Score
STAS (absent, present)	0–1
Histology (lepidic, acinar, solid and other) [20]	1–2–3
Grading (G1, G2, G3)	1–2–3
Necrosis (≤10%, 11–30%, >30%)	0–1–2
Mitoses (≤10%, 11–30%, >30%)	0–1–2
TILs (<11%, ≥11%)	0–1
Fibrosis (≤10%, 11–30%, >30%)	0–1–2
Vascular invasion (absent, present)	0–1
Perineural invasion (absent, present)	0–1
Pleural invasion (PL0, PL1, PL2)	0–1–2
TOTAL	18 Points

**Table 2 cancers-15-04542-t002:** Perioperative data.

Variables (*n* = 358) Median (IQR); *n* (%)	N = 358
**Age at surgery**	70 (63, 75)
Gender	
Female	179 (50%)
Male	179 (50%)
**BMI**	26.2 (22.9, 28.7)
**Diabetes**	38 (11%)
**Hypertension**	206 (58%)
**COPD**	43 (12%)
**Smoking history**	
No	99 (28%)
Active	221 (62%)
Former (at least one month)	38 (11%)
**FVC%**	100 (88, 113)
**FEV1%**	100 (84, 113)
**DLCO/VA%**	77 (64, 90)
**Surgical time (min)**	120 (90, 150)
**Typical resection type**	
Segmentectomy	78 (22%)
Lobectomy	280 (78%)
**Procedure Access**	
Uniportal VATS	6 (1.7%)
Biportal VATS	28 (7.8%)
Triportal VATS—Copenhagen	319 (89%)
Robotic	4 (1.1%)
**Lobectomies**	
RUL	121 (34%)
ML	25 (7.0%)
RLL	50 (14%)
LUL	57 (16%)
LLL	27 (8%)
**Side**	
Right	223 (62%)
Left	135 (38%)
**Conversion rate**	4 (1.1%)
**cT diameter (mm)**	20 (13, 29)
**cT**	
1a	28 (7.8%)
1b	136 (38%)
1c	117 (33%)
2a	49 (14%)
2b	28 (7.8%)
**Peritumoral GGO**	123 (34)
**SUV T**	3.7 (2.0, 6.7)
**Preoperative diagnosis**	180 (50%)

Footnote. BMI: Body Mass Index, COPD: Chronic Obstructive Pulmonary Disease, FEV1%: Forced Expiratory Volume in 1 s (percentage), FVC%: Forced Vital Capacity (percentage), DLCO/VA%: blood transfer coefficient for the diffusion of CO (percentage). RUL: right upper lobe, ML: middle lobe, RLL: right lower lobe, LUL: left upper lobe, LLL: left lower lobe.

**Table 3 cancers-15-04542-t003:** Pathological data available into the pathological report.

Pathological Characteristics (n = 358)	
**pT**	
1a	54 (15%)
1b	103 (29%)
1c	28 (7.8%)
2a	159 (44%)
2b	14 (4%)
**p-Stage**	
IA1	62 (17%)
IA2	83 (23%)
IA3	22 (6.1%)
IB	176 (49%)
IIA	15 (4.2%)
**STAS type**	
Absent	207 (58%)
Limited	135 (38%)
Extensive	12 (3.4%)
**Histology**	
Lepidic	41 (11%)
Acinar/papillar	279 (78%)
Solid	38 (11%)
**Grading**	
G1	11 (3.1%)
G2	258 (73%)
G3	79 (22%)
**Tumor necrosis**	
<10%	305 (86%)
11–30%	32 (9.1%)
>30%	16 (4.5%)
**Mitoses**	
0–1/10HPF	132 (54%)
2–4/10HPF	83 (34%)
>4/10HPF	30 (12%)
**TILs**	
<11%	142 (40%)
11–30%	158 (45%)
>30%	54 (15%)
**Fibrosis**	
<10%	164 (49%)
11–30%	102 (31%)
>30%	68 (20%)
**Vascular invasion**	
Absent	290 (82%)
Present	65 (18%)
**Perineural invasion**	
Absent	270 (98%)
Present	5 (1.8%)
**Pleural invasion**	
PL0	141 (42%)
PL1	175 (52%)
PL2	19 (5.7%)
**PD-L1**	
<1%	156 (53%)
≥1%	140 (47%)
**Median Surveillance in months**	34 (18–55)

Footnote. HPF: high-power field; PL: Pleural invasion (0 = Neoplasm-free pleura, 1 = Limited visceral pleura involvement (no involvement of elastic part), 2 = Limited visceral pleura involvement (beyond the elastic part); PD-L1: Programmed Death Ligand 1).

**Table 4 cancers-15-04542-t004:** Pathological comparisons according to the PD-L1 expression (<1% or ≥1%).

Characteristic	N	PD-L1 < 1% N = 156 (53%)	PD-L1 almost 1% N = 140 (47%)	*p*-Value
**SUV T**	226	3.1 (1.7, 5.3)	4.0 (2.4, 7.9)	**0.010**
**STAS**	294			0.26
Absent		88 (57%)	89 (64%)	
Present		66 (43%)	51 (36%)	
**Histology**	296			0.26
Lepidic		20 (13%)	13 (9.3%)	
Acinar/papillar		123 (79%)	108 (77%)	
Solid		13 (8.3%)	19 (14%)	
**Grading**	291			0.33
G1		7 (4.6%)	2 (1.4%)	
G2		112 (73%)	98 (71%)	
G3		31 (20%)	36 (26%)	
**Tumor necrosis**	296			0.17
<10%		140 (90%)	117 (84%)	
11–30%		10 (6.4%)	18 (13%)	
>30%		6 (3.8%)	5 (3.6%)	
**Mitoses**	296			0.95
0–1/10HPF		106 (68%)	97 (69%)	
2–4/10HPF		38 (24%)	32 (23%)	
>4/10HPF		12 (7.7%)	11 (7.9%)	
**Fibrosis**	296			0.30
<10%		88 (56%)	70 (50%)	
11–30%		46 (29%)	41 (29%)	
>30%		22 (14%)	29 (21%)	
**Vascular invasion**	296			0.19
Absent		126 (81%)	121 (86%)	
Present		30 (19%)	19 (14%)	
**Perineural invasion**	296			>0.99
Absent		154 (99%)	139 (99%)	
Present		2 (1.3%)	1 (0.7%)	
**Pleural invasion**	296			0.57
PL0		74 (47%)	59 (42%)	
PL1		74 (47%)	75 (54%)	
PL2		8 (5.1%)	6 (4.3%)	
**TILs**	292			0.12
<11%		70 (46%)	51 (37%)	
≥11%		83 (54%)	88 (63%)	
Comparisons: Wilcoxon rank sum test; Pearson’s chi-squared test; Fisher’s exact test				

## Data Availability

Not applicable.

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
