# Peer review of "Microscopical Variables and Tumor Inflammatory Microenvironment Do Not Modify Survival or Recurrence in Stage I-IIA Lung Adenocarcinomas"

_cancers, 2023, doi:10.3390/cancers15184542_

Round 1

Reviewer 1 Report

 The article is well-organized, with clearly defined sections that effectively convey the main arguments. However, there are a few minor issues that require attention and clarification. Firstly, it would be helpful to provide information about the time unit used in the x-axis of the charts presented in Figures 3, 4, 5, and 6. This would enable readers to better understand the temporal trends being depicted and put the data into context. Secondly, the caption for Figure 4 is inappropriate and needs to be modified further."

When faced with complex sentences that are challenging to comprehend or contain grammatical errors, it is crucial to conduct a thorough review and implement necessary modifications. By doing so, we can enhance the clarity and readability of these sentences, ensuring they effectively convey the intended message.

Author Response

Dear Editors and Reviewers, 
Dear Editorial Office 

Thank you very much for your revision on our article entitled: “Microscopical 
variables and tumor microenvironment do not modify survival or recurrence in 
stage I-IIA lung adenocarcinomas.”. 

We attached a pdf version of our response letter below. 

Thank you for your consideration of this manuscript. 

Sincerely, 

Prof. Andrea Dell’Amore 

[email protected] 
Thoracic Surgery Unit, Department of Cardiac, Thoracic and Vascular Sciences, 
University of Padova 
Via Giustiniani 2, 35128, Padova, Italy

Reviewer 2 Report

Correlation with imaging data would also be an important part, would you consider adding those? Overall interesting work on the data regarding tumour microenvironment. Would you have any correlation with Cancer associated fibroblasts (CAFS) in your specimen sample, to assess further the tumour infiltration in surgical margins or the effect of the microenvironment overall on your specimen? Also any ctDNA data to corroborate your findings? 

Author Response

Dear Editors and Reviewers, 
Dear Editorial Office 

Thank you very much for your revision on our article entitled: “Microscopical 
variables and tumor microenvironment do not modify survival or recurrence in 
stage I-IIA lung adenocarcinomas.”. 

We attached a pdf version of our response below. 

Thank you for your consideration of this manuscript. 

Sincerely, 

Prof. Andrea Dell’Amore 

[email protected] 
Thoracic Surgery Unit, Department of Cardiac, Thoracic and Vascular Sciences, 
University of Padova 
Via Giustiniani 2, 35128, Padova, Italy

Reviewer 3 Report

Overall, the manuscript is well-structured and well-written. Moreover, it is interesting from the clinician's point of view. Its originality is in the focus of authors on low-stage cancers (I and II), that are usually less studied since surgery is the gold standard.

Although, in my opinion, the title might be misleading, since I was expected to find a deepened analysis of the tumour microenvironment, that I could not find along the manuscript.  

Minor English revision.

Author Response

(The authors gave the same response as above.)
